# *Drosophila* as a Model for Human Disease: Insights into Rare and Ultra-Rare Diseases

**DOI:** 10.3390/insects15110870

**Published:** 2024-11-06

**Authors:** Sergio Casas-Tintó

**Affiliations:** Institute for Rare Diseases Research, Instituto de Salud Carlos III (ISCIII), 28222 Madrid, Spain; sergio.casas@isciii.es

**Keywords:** genetics, neuroscience, metabolism, cancer, animal model, development, CRISPR, *Drosophila melanogaster*, avatars

## Abstract

Rare and ultra-rare diseases are challenging to study and treat because we do not fully understand their causes or how they work. These diseases often show a wide variety of symptoms and complex genetic factors, making research difficult. There are over 7000 different rare diseases affecting nearly 300 million people worldwide, but because each disease affects only a small number of people, the resources dedicated to studying each one effectively are a limiting factor. To help with research, scientists use animal models to explore how these diseases develop. One of the most useful models is the fruit fly, *Drosophila melanogaster*. This popular model in research has a short life cycle, is easy to maintain in a lab, and has a fully mapped genome. Moreover, many of its genes are similar or equivalent to those in humans, making it a valuable tool for studying genetic disorders and initiating the development of treatments for rare diseases. This text looks at how *Drosophila melanogaster* is used in research on rare diseases, emphasizing its importance and potential in advancing biomedical research.

## 1. What Is a Rare and Ultra-Rare Disease?

Rare diseases are defined as medical conditions that impact a small percentage of the population. The criteria for classification can vary by country; for instance, the European Commission defines a disease as rare if it affects fewer than 1 in 2000 individuals within a given population [1], and in the United States, a rare disease is one that affects fewer than 200,000 people, according to the NIH (www.nih.gov) (accessed on 20 October 2024). In certain cases, diseases with a prevalence lower than 1 in 50,000 individuals are categorized as ultra-rare diseases (Table 1).

According to the European Medicines Agency (EMA), there are between 5000 and 8000 known rare diseases. Even though individual rare diseases are uncommon, collectively rare diseases affect around 300 million of people worldwide. Some rare diseases, like cystic fibrosis or Huntington’s disease, affect larger populations, while others are extremely infrequent. These numbers fluctuate as new diseases are discovered and more information becomes available through research and improved diagnostics. Since each rare disease affects a small number of people, the resources dedicated to each of these conditions are often limited compared to more prevalent diseases.

Due to the limited understanding and resources available for rare and ultra-rare diseases, they often present unique challenges for patients, healthcare providers, and researchers. To obtain an accurate diagnosis and access to appropriate treatments and clinical trials is particularly difficult for these patients [2,3]. Thus, the development of novel research strategies, such as *Drosophila* models of disease [4], minimize the use of resources and maximize the results required to develop targeted therapies for rare and ultra-rare diseases (Table 1).

**Table 1 insects-15-00870-t001:** Examples of human rare and ultra-rare diseases modelled in *Drosophila melanogaster*.

Disease	Gene (Human)	Gene (*Drosophila*)	Publication
Phosphatidylinositol glycan biosynthesis class A congenital disorder of glycosylation (PIGA-CDG)-seizures, intellectual and developmental delay, and congenital malformations	*phosphatidylinositol glycan biosynthesis class A* (*PIGA*)	*PIG-A*	*Drosophila* models of phosphatidylinositol glycan biosynthesis class A congenital disorder of glycosylation (PIGA-CDG) mirror patient phenotypes [5].
Developmental and epileptic encephalopathy 44 (DEE44)-early-onset encephalopathy, movement abnormalities, global developmental delay, intellectual disability, and seizures	*Ubiquitin-like modifier activating enzyme 5* (*UBA5*)	*Uba5*	Allelic strengths of encephalopathy-associated *UBA5* variants correlate between in vivo and in vitro assays [6].
Developmental and epileptic encephalopathy (DEE)	*Potassium voltage-gated channel subfamily A member 3* (*KCNA3*)	*Shaker* (*Sh*)	De novo variants in KCNA3 cause developmental and epileptic encephalopathy [7].
Myotonic dystrophy type 1 (DM1)-progressive muscle dysfunction, weakness, myotonia, and wasting	*muscleblind-like family factors* (*MBNL1-MBNL3*	*Muscleblind-like* (*MBNL*)	Development of a *Drosophila melanogaster* spliceosensor system for in vivo high-throughput screening in myotonic dystrophy type 1 [8]. Natural compound boldine lessens Myotonic Dystrophy Type 1 phenotypes in DM1 *Drosophila* models, patient-derived cell lines, and HSA^LR^ mice [9].
X-linked syndromic intellectual developmental disorder-34 (MRXS34)-Developmental delay, distinctive facial features, cardiac symptoms, and skeletal symptoms	*NONO*	*nonA*	A novel *NONO* variant that causes developmental delay and cardiac phenotypes [10].
Fatty acid hydroxylase-associated neurodegeneration (FAHN)-brain modifications and motor dysfunctions in early childhood	*Fatty acid 2 hydroxylase* (*FA2H*)	*dfa2h*	A new model for fatty acid hydroxylase-associated neurodegeneration reveals mitochondrial and autophagy abnormalities [11].
NGLY1 deficiency-global developmental delay, movement disorders, seizures, orthopaedic manifestations, microcephaly, feeding difficulty, chronic constipation, elevated liver enzymes, and intellectual disability	*N-glycanase 1* (*NGLY1*)	*PNGase-like* (*Pngl*)	Tracing the NGLY1 footprints: insights from *Drosophila* [12].
**Unnamed Disease**	**Gene (Human)**	**Gene (*Drosophila*)**	**Publication**
Hypotonia, global developmental delay, epileptic encephalopathy, and dysmorphic features	*Cyclin-dependent kinase 19* (*CDK19*)	*Cdk8*	De novo variants in CDK19 are associated with a syndrome involving intellectual disability and epileptic encephalopathy [13].
Global developmental delay (GDD), dysmorphic features, ophthalmologic abnormalities, and neurological features	*Transportin-2* (*TNPO2*)	*Transportin* (*dTnpo*)	TNPO2 variants associate with human developmental delays, neurologic deficits, and dysmorphic features and alter TNPO2 activity in *Drosophila* [14].
Developmental and language delay, intellectual disability, impaired motor development, and brain atrophy	*Integrator subunit* (*11INTS11*)	*Integrator 11* (*dIntS11*)	Bi-allelic variants in INTS11 are associated with a complex neurological disorder [15].
Neurodevelopmental disorder (NDD)-developmental delay and intellectual disability (DD/ID), hypotonia, and neurobehavioral problems, with variable skeletal (66.7%) and cardiac (46%) anomalies	*non-small nuclear ribonucleoprotein* (*non-snRNP*) *that belongs to the arginine/serine* (*R/S*) *domain family* (*SRSF1* (*also known as ASF/SF2*))	*Splicing factor 2* (*SF2*)	SRSF1 haploinsufficiency is responsible for a syndromic developmental disorder associated with intellectual disability [16].
Developmental delay, intellectual disability, dysmorphic features, and other congenital anomalies	*FRY-like transcription coactivator* (*FRYL*)	*furry* (*fry*)	De novo variants in FRYL are associated with developmental delay, intellectual disability, and dysmorphic features [17].

Source orpha.net; marrvel.org.

## 2. Genetics of Rare Diseases

The genetic basis of rare diseases varies widely depending on the specific condition. Most rare diseases are caused by genetic mutations, which are alterations in the DNA sequence of genes. These mutations can be inherited from parents or de novo mutations that arise spontaneously. The nature of each mutation can be different and, therefore, the mechanisms underlying the disease are different from patient to patient. There are different types of genetic alterations that can cause a rare disease (Figure 1). It is highly frequent that a rare disease is caused by a mutation in a single gene. These mutations can be inherited from one or both parents (inherited rare diseases) or occur spontaneously during the development of an individual (de novo mutations). Examples include cystic fibrosis, Duchenne muscular dystrophy, and Huntington’s disease [18,19,20]. Also, mutations that alter the number of copies of a particular segment of DNA, such as duplications or deletions of gene regions, are pathological. These genetic alterations are known as copy number variations (CNVs). For instance, a deletion in the chromosome 15q11.2 region can lead to a rare neurodevelopmental disorder called 15q11.2 deletion syndrome [21,22]. In a larger scale, changes in the structure or number of chromosomes that cause chromosomal abnormalities affect the expression of multiple genes. For example Down syndrome is caused by the presence of an extra copy of chromosome 21 [23,24], and Turner syndrome is caused by the partial or complete absence of one X sex chromosome [25].

In addition, mutations in mitochondrial genes lead to rare mitochondrial diseases, which can affect various tissues due to the central role of mitochondria in cell function [26]. Mitochondria have their own DNA (mtDNA), which only contains 37 genes [27]. Some of the most common mitochondrial diseases include Leigh syndrome, caused by mutations in mitochondrial genes involved in the oxidative phosphorylation chain [28], or MELAS (Mitochondrial Encephalopathy, Lactic Acidosis, and Stroke-like episodes), also caused by mitochondrial mutations [29].

It is generally accepted that mammalian early embryos eliminate paternal mitochondria according to the “embryo silent” hypothesis, and in consequence, the maternal contribution of mitochondria is larger if not unique [30]. Thus, it is considered that mitochondrial diseases can be only maternally inherited [31]. However, recent discoveries suggest that a minor burden of mitochondrial inheritance may come from the father, opening an interesting debate [32].

Certain rare diseases involve multiple genetic variants across different genes; these variations are known as polygenic inheritance. This complex situation is worsened by the fact that the individual contribution of each gene might be unnoticeable and, therefore, only the combination of mutations is pathological [33]. The combined effects of mutations in multiple genes and the influence of environmental factors result in complex traits that cause rare and ultra-rare diseases including cardiovascular affections such as coronary artery disease and atrial fibrillation [34], and neurodevelopmental disorders such as autism or schizophrenia [35].

These genetic alterations can be reproduced in research models if the nucleotide, gene, or genomic region is conserved from flies to humans. Thus, genetic and functional conservation is a major requirement for modelling genetic diseases in *Drosophila.* Point mutations, CNV, and mitochondrial mutations imply smaller regions of DNA and, therefore, are easier to reproduce. More complex scenarios including chromosomal abnormalities or polygenic inheritance are less feasible and usually represent a limitation for animal model generation.

The genetic basis of some rare diseases remains unknown and the advances in genetic technologies and research, such as whole-genome sequencing and functional genomics, bring novel opportunities to understand the genetic causes underlying rare diseases. This knowledge is crucial for diagnosis, prognosis, and the development of targeted therapies but requires the functional validation of these mutations in undiagnosed individuals.

## 3. *Drosophila* Conservation

*Drosophila melanogaster* and humans share 60% of their genetic conservation, and up to 75% of the genes associated with human disease have an orthologue in *Drosophila melanogaster* [36,37,38,39,40,41]. Specialized tools can predict orthologue genes by sequence, expression pattern, and function. Upon the comparison of human and fly genomes, these tools can determine the degree of conservation. DIOPT (https://www.flyrnai.org/diopt) (accessed on 10 October 2024) integrates human, mouse, fly, worm, zebrafish, and yeast genomes and allows predictions based on sequence homology, phylogenetic trees, and functional similarity. There are more specialized resources to search for the conservation of specific rare disease variants and the conservation through different species, including humans and *Drosophila melanogaster*. MARRVEL is the Model Organism Aggregated Resources for Rare Varian ExpLoration (https://marrvel.org/) (accessed on 10 October 2024). This Web-based tool integrates human genomics and model organism genetic information, which allows the bioinformatic analysis of human gene variants’ pathogenicity and the conservation through species in sequence, expression of orthologue genes in tissues, phenotypes, gene ontology and protein sequence alignment, as well as further information on protein structure prediction and drugs and active ligands for the diseases associated to mutation in specific genes.

Orthologous genes often perform similar functions in both species, indicating the evolutionary conservation of gene function. The fundamental genetic pathways controlling development are highly conserved between *Drosophila* and humans and several key players in development were initially described in flies. One of the best studied cases is the gene called *eyeless*, an orthologue to the human gene *Pax6* that is essential for eye development in both organisms [42]. Also, the *HOX* genes, which regulate body segment identity during embryonic development, are present in both species and have similar functions [43].

Other conserved pathways include those involved in cell cycle regulation, with the discovery of the cyclin-dependent kinase (CDK), Cyclin E, Cyclin A, and Cyclin B [43], signaling pathways as hedgehog [44,45], Notch [46], Wingless (WNT) [47,48], Dpp (transforming growth factor-beta in humans) [49], and EGFR [50,51]. These pathways are often involved in human disease [51,52,53,54,55], and in consequence, *Drosophila* emerges as a suitable model to reproduce in vivo scenarios to reproduce pathological alterations in these pathways.

Moreover, specific human disease-related genes have counterparts in *Drosophila* that contribute to similar disease-like phenotypes. For example, the gene responsible for cystic fibrosis in humans, *CFTCR* (*Cystic Fibrosis Transmembrane Conductance Regulator*), has a counterpart in *Drosophila* called *dCFTR*. Mutations in *dCFTR* produce similar symptoms in *Drosophila*, indicating that it is a useful model for studying aspects of cystic fibrosis [18].

While there are notable differences between *Drosophila* and humans, the genetic conservation between the two species has allowed researchers to gain significant insights into basic biological processes, disease mechanisms, and potential therapeutic targets. The use of *Drosophila* as a model organism has proven valuable in advancing our understanding of human genetics and diseases.

## 4. Benefits of Using *Drosophila* as Model Organism

One of the most relevant advantages of *Drosophila* is the collection of genetic tools for genetic research, and many genetic tools and techniques developed in *Drosophila* have been adapted for use in mammalian research. Among them are the technique to introduce transgenes into the genome using pronuclear injection or viral-mediated transgenesis [56,57], the Gal4/UAS system originally developed in *Drosophila* from yeast [58], other binary expression systems including *LexA/LexAop* and *QF/QUAS* [59,60], the RNA of interference (RNAi), and the Flp/FRT system of site-specific recombination [61,62]. Although CRISPR-Cas9 system was not initially developed in *Drosophila*, it was largely adapted to flies and then optimized for mammalian research. The versatility of these techniques is an example of how *Drosophila* genetics has contributed to our understanding of gene function and development in biology, including human biology and diseases.

Binary expression systems (*Gal4/UAS*, *LexA/LexAop*, *QF/QUAS*) [58,59,60] operate on a common framework comprising a transcription factor (TF) and a specific DNA sequence recognized by it. These systems express the TF (*Gal4*, *LexA* or *QS*) under the control of a specific promoter that determines the expression pattern. Each TF binds to the corresponding sequence (*UAS*, *LexAop* or *QUAS*) and promotes the expression of the gene downstream. In consequence, the gene(s) of interest or the *RNAi* are expressed in a specific pattern of interest. These genetic tools are the base of the vast array of transgenic flies that, along with CRISPR/Cas9, clonal analysis, RNAi techniques use to facilitate the use of *Drosophila* as an ideal organism for genetic studies.

### CRISPR/Cas9 System to Create Drosophila Avatars

The genetic tools utilized in *Drosophila* research contribute to the “humanization” of fly genes to reproduce human pathological mutations, study rare variants in human proteins, perform phenotypic analysis in specific cells or tissues, or study truncated proteins during development or at later stages [4,63]. Among them, the CRISPR/Cas9 system has recently emerged as a pivotal technique for inducing targeted genomic modifications, including insertions, deletions, and point mutations. In vivo precise genome editing provides a unique tool to reproduce mutations with minimal alteration of the genome [64].

The CRISPR/Cas9 adaptative immune system from *S. pyogenes* was simplified and adapted to modify eucaryotic genomes. The system requires the Cas9 protein in germ line cells to produce mutant flies, or in specific tissues of the animal. This technique takes advantage of the binary expression systems, mostly Gal4/UAS, to drive the expression of the *Cas9* gene under the control of specific promoters. In addition, specific guide RNA directs the site cleavage of genomic DNA by Cas9. These guides recognize a 20-nt target sequence that, recognized by Cas9, undergoes double strand breaks (DSBs).

DSBs induce DNA repair mechanisms that are used to edit the genome at this locus. There are two repair systems, nonhomologous en jointing (NHEJ) and homology-directed repair (HDR); the last is a precise system of repair that employs DNA sequences as templates. By providing an external donor template, this repair pathway can be harnessed to accurately modify genomic sequences or introduce foreign DNA [64,65].

Each component of the CRISPR/Cas9 system has to be present in the same cell to produce the reaction. There are several possibilities from injecting all components in the embryo, or on the contrary, to generate stable transgenic flies and cross them (reviewed in [65]). In our hands, to generate rare disease avatars, our aim is to reproduce point mutations in conserved genes; thus, the optimal strategy is to inject a DNA plasmid containing the gRNA and the specific donor sequences in embryos from a transgenic line expressing *Cas9* in the germ line (Figure 2).

This capability allows the recreation of a wide array of mutations observed in human patients. Furthermore, the ability to generate functional assays using *Drosophila* models offers significant advantages for undiagnosed patients with suspected pathogenic loci, enabling the investigation of novel gene–disease associations and the confirmation of the pathogenic role of specific genetic variants [4,66]. Most rare diseases affect developmental processes during the early stages of life and, therefore, knowledge of the genes that direct the cells to form a body during development is of relevance. *Drosophila* has a well-characterized and highly conserved process of embryonic development despite evident morphological and anatomical differences with humans. The heyday of the use of *Drosophila* as a model to study developmental biology occurred after the work of Nusslein-Volhard and Wieschaus on the organization of the body plan and the segmental pattern during development [67], which led to a Nobel prize in 1995. Research in the fly provides insights into the fundamental mechanisms underlying developmental processes, such as cell differentiation, pattern formation, and organogenesis [47,68,69], and has provided a basis for understanding human development and diseases [70,71,72].

## 5. Human Diseases Modelled in *Drosophila*

The use of *Drosophila melanogaster* as a model organism in biomedical research has provided crucial insights into the genetic and molecular mechanisms of the most prevalent human diseases. Due to its genetic manipulability, relatively simple biology, and conservation of key pathways, *Drosophila* offers a valuable system to model a wide range of human diseases, from muscular and neurodegenerative disorders to cancer. This section explores how researchers have leveraged *Drosophila* to study specific disease types, including muscular dystrophies, neurodegenerative conditions like Alzheimer’s and Parkinson’s, and glioblastoma as examples. Through these models, researchers dissect disease mechanisms, identify genetic modifiers, and screen for potential therapeutic strategies, making *Drosophila* an indispensable tool in advancing our understanding of complex human diseases.

### 5.1. Muscular Diseases

*Drosophila* is a suitable model organism to study muscular diseases, particularly those affecting muscle development, function, and degeneration. *Drosophila* models offer several advantages for studying muscular diseases, including the structural conservation of muscular tissues structure, genetic manipulability, and the ability to perform high-throughput screens.

Muscular dystrophies are a group of diseases that affect muscular tissues or cells. Muscle cells progressively degenerate, and muscle weakness decreases mobility and severely affects the mobility of patients. *Drosophila* models have contributed to our understanding of various forms of muscular dystrophy, including Duchenne muscular dystrophy (DMD). For example, *Drosophila* models expressing a mutated form of the *dystrophin* gene (the gene responsible for DMD) reproduce the muscle degeneration, impaired locomotion, and compromised muscle function observed in patients, and have been used to study the molecular mechanisms underlying muscle degeneration, inflammation, and potential therapeutic interventions [19].

Myotonic dystrophy is a genetic disorder characterized by muscle weakness and myotonia. *Drosophila* models based on the expression of expanded repeat RNA sequences associated with DM1 or DM2 lead to muscle defects, impaired flight, and altered splicing patterns reminiscent of human diseases [73,74,75].

*Drosophila* models also reproduce certain forms of cardiomyopathy, which can involve impaired heart muscle function. The manipulation of genes associated with cardiomyopathy in *Drosophila* lead to abnormal heart morphology, compromised cardiac function, and reduced lifespan [76,77].

These models have provided valuable insights into the genetic and molecular mechanisms underlying muscular diseases and have facilitated the development of potential treatments and therapeutic strategies [78,79].

### 5.2. Neurodegenerative Diseases

*Drosophila* is widely used as a model organism to study neurodegenerative diseases. Here are a few examples of how *Drosophila* has contributed to our understanding of these maladies:

Alzheimer’s disease (AD) is a progressive and irreversible neurological disorder that primarily affects the brain, leading to a gradual decline in memory, thinking skills, and cognitive function. *Drosophila* models have been instrumental in studying the genetic and molecular mechanisms underlying Alzheimer’s disease [52]. Expression of human *amyloid beta* (*Aβ*) forms, a key pathological hallmark of AD, in the fly brain leads to the formation of Aβ aggregates and cognitive deficits [52,80]. Researchers have used *Drosophila* to identify genetic modifiers, signaling pathways, and potential therapeutic targets involved in AD progression [81].

Parkinson’s disease (PD) is a progressive neurological disorder that primarily affects movement. It is characterized by a loss of dopamine-producing cells in a region of the brain called the substantia nigra. Dopamine is a chemical messenger responsible for transmitting signals that control movement and coordination. *Drosophila* models have provided valuable insights into the pathogenesis of PD. Expression of mutant forms of human genes such as *alpha-synuclein*, *parkin*, and *PINK1* in *Drosophila* recapitulates key features of PD, including dopaminergic neuron loss and motor deficits. These models have been instrumental in elucidating cellular and molecular mechanisms, identifying genetic modifiers, and testing potential therapeutic strategies for Parkinson’s disease [82,83].

Huntington’s disease (HD) is a hereditary neurodegenerative disorder characterized by the progressive loss of neurons in the brain, leading to a wide range of physical, cognitive, and psychiatric symptoms. Huntington’s disease is caused by a mutation in the *huntingtin* (*HTT*) gene, which leads to the production of an abnormal form of the huntingtin protein. This mutated protein gradually accumulates in the brain, particularly in the basal ganglia and cerebral cortex, causing damage to neurons and ultimately resulting in the characteristic symptoms of the disease. *Drosophila* models expressing mutant forms of the huntingtin protein, responsible for HD, have been used to study the molecular and cellular consequences of mutant huntingtin toxicity [84,85]. These models have shed light on the mechanisms underlying neurodegeneration, mitochondrial dysfunction, transcriptional dysregulation, and oxidative stress in Huntington’s disease [20].

The *Drosophila* Amyotrophic Lateral Sclerosis (ALS) model expresses mutant forms of genes associated with ALS, such as *superoxide dismutase 1* (*SOD1*) and *TAR DNA-binding protein 43* (*TDP-43*), have contributed to our understanding of ALS pathogenesis. *Drosophila* models have been used to investigate mechanisms of motor neuron degeneration, protein aggregation, impaired axonal transport, and glial dysfunction in ALS [86,87].

A final example is Spinocerebellar Ataxias (SCAs). Drosophila models expressing mutant forms of various SCAs, including ataxin-1 [88], ataxin-3 [89], and ataxin-7 [90], have helped to elucidate the role of protein aggregation, impaired protein degradation, altered gene expression, and mitochondrial dysfunction in SCA pathogenesis.

These examples demonstrate how *Drosophila* models have contributed to our understanding of the genetic and molecular mechanisms underlying neurodegenerative diseases. The relative simplicity of the fly nervous system, ease of genetic manipulation, and the ability to perform high-throughput screens make *Drosophila* an invaluable tool for studying these complex disorders.

### 5.3. Drosophila melanogaster in Glioblastoma Research

One of the greatest challenges in the last decade of biomedical research is the cure of glioblastoma (GB). This is the most aggressive form of primary brain cancer, with a median survival of 15 months after diagnosis [91].

GB is characterized by high invasiveness, rapid progression, and resistance to therapy. Understanding the molecular and cellular mechanisms underlying its complex biology is a challenge, and developing effective therapies is an urgent need. Due to the limited availability of human tissue samples and the complex interactions within the brain microenvironment, model organisms offer valuable platforms for investigating glioblastoma pathogenesis. *Drosophila* has emerged as a valuable model organism in the field [92,93]. Recent advances in GB biology have been discovered and/or validated in flies, including the contribution to the cell-to-cell communication of the WNT [54], insulin [94], and JNK [95] pathways, the formation of neuron-GB and intratumoral synapses [96], the impact of GB on circadian rhythms [97], the vulnerability of GB to YAP/TAZ pathway activation [98], and the contribution of vesicular transport to GB progression [53,99].

Ongoing advancements in *Drosophila* research and disease biology present exciting opportunities for future investigations. The integration of multi-omics approaches, including genomics, transcriptomics, and proteomics, combined with mathematical models and computational analyses, will enhance our understanding of the complex molecular networks underlying disease mechanisms and open novel opportunities for personalized medicine as a standard of care.

## 6. *Drosophila* in Rare Disease Research

Rare diseases pose unique challenges in terms of understanding their underlying mechanisms and developing effective treatments. The use of model organisms, especially *Drosophila*, has contributed to rare disease research and provides platforms for therapeutic target identification.

In the case of rare diseases, the low maintenance costs enable efficient experimental designs to fit with the short budget dedicated to each specific rare and ultra-rare disease. Furthermore, as we discussed above, the fly shares conserved signaling pathways, cellular processes, and genetic elements with humans, facilitating the translation of findings to human disease contexts. In addition, the generation of functional studies and high throughput screening, and the limited ethical restrictions that apply to this model organism, make *Drosophila* particularly suitable for the study of genetic rare disease.

The utilization of flies has yielded significant insights into various rare diseases. By leveraging its genetic tools, researchers have successfully modelled and characterized the pathogenic mechanisms of specific disorders [100]. The fly has contributed significantly in identifying disease-causing genes, elucidating molecular pathways, and establishing genotype–phenotype correlations [101,102,103].

Moreover, large-scale genetic screens in *Drosophila* have unveiled novel candidate genes involved in rare diseases, broadening our understanding of their genetic landscape [102,104,105]. Techniques such as RNA interference, transgenesis, and CRISPR-Cas9 genome editing enable the precise manipulation of gene expression, mimicking disease-associated genetic alterations and facilitating personalized medicine strategies [106,107].

Recent strategies focused on the generation of genetic avatars aim to reproduce, with nucleotide precision, the mutations found in patients in *Drosophila*. The term avatar is used to describe a model that reproduces a human condition or phenotype. *Drosophila* avatars are a powerful tool in biomedical research, particularly in the study of genetic diseases and disorders (Figure 2). Due to the genetic similarities and conserved biological processes between flies and humans, these genetic models bring valuable insights into human diseases. This strategy creates a new pathway toward personalized medicine by enabling treatment testing on avatars [105,108]. Additionally, sophisticated imaging [109], behavioral assays [110,111], and physiological measurements [109,112] in flies provide insights into disease phenotypes and their functional consequences.

The emerging trends in the field and the prospects rely on the latest advancements in genomic technologies, such as high-throughput sequencing and functional genomics, that present exciting opportunities for rare diseases research in *Drosophila*. The integration of multi-omics datasets, network analyses, and advanced imaging techniques will enhance our understanding of disease mechanisms and aid in the identification of novel therapeutic targets. Furthermore, the integration of patient-derived data and *Drosophila* models hold promise for personalized medicine approaches in rare diseases.

The primary applications of *Drosophila* in biomedical research cover key aspects modelling human rare diseases, where genetic modifications play a crucial role as they represent the main cause of such conditions. Rare diseases manifest during developmental stages, and *Drosophila* provides a unique platform to reproduce features of human disease and develop genetic or drug screenings as an initial step in biomedical research [100].

## 7. Limitations of Using *Drosophila* to Model Disease

The use of animal models presents limitations that are of great relevance for biomedical research and the extrapolation of the results to human disease. Rare diseases often compromise neurological symptoms that cause behavioral alterations. Non-human organisms often lack the complex cognitive abilities found in humans, and therefore, the analysis of behavioral aspects in model organisms is a limitation, and in *Drosophila* it is limited to relatively simple behaviors including learning and memory paradigms [113]. Although these experimental paradigms are solid and reliable, there is a need for improved methods to measure and understand behavioral tendencies in non-human organisms to better transfer the results to humans. This would contribute to gaining insights into the cognitive processes and decision-making abilities.

Even though the genetic conservation between *Drosophila* and humans is high, around 40% of genes present in humans do not exist in the fly or have a different function. This is a great and significant limitation that must be taken into consideration for biomedical research. To work around these limitations, the introduction of human genes in flies has partially provided a solution. Human disease-related genes exert a comparable effect in *Drosophila* cells, as, for example, Abeta42 in Alzheimer’s disease models [114]. *Drosophila* contains an *amyloid precursor protein-like* gene [115], but this protein does not produce the peptide Abeta42 responsible for AD progression. The insertion of human Abeta42 sequence produces Abeta42 peptide and the flies display progressive neurodegeneration [52], as in humans.

In addition, *Drosophila* does not possess an adaptive immune system like that of humans. This can present challenges when studying immune responses and developing treatments or vaccines. For example, advanced therapies such as CAR-T involve the genetic modification of immune cells and flies are not a suitable model for these studies. More specific studies on the sensitivity of *Drosophila* to human pathogenic and non-pathogenic gram-positive bacteria showed that it is not a suitable model, as non-pathogenic gram-positive bacteria results in high mortality [116].

Anatomical differences between *Drosophila* and humans are evident, and this might result in a limitation for the study of specific tissues such as the circulatory system. *Drosophila* has an open circulatory system with a simple heart that pumps the haemolymph. In addition, it has a tracheal system that reproduces, partially, the cellular and molecular properties of the circulatory system [117].

Lastly, the effects of drugs and therapeutic interventions can differ significantly between non-human organisms and humans. This variation can pose challenges when attempting to extrapolate findings from non-human studies to human populations, potentially leading to unexpected outcomes and limited applicability.

Addressing these challenges requires continued research and the development of innovative methods and approaches to improve our understanding of non-human organisms, their behaviors, cognitive abilities, and immune systems, and the translation of research findings to human contexts. Also, it is of great relevance to consider the limitations of *Drosophila* and other model organisms to optimize the design of the strategies to study human disease.

*Drosophila melanogaster* is a highly effective and valuable research model for studying rare and ultra-rare diseases. Its amenability to genetic manipulation, conservation of key molecular pathways, and ability to recapitulate disease phenotypes make it an indispensable tool in deciphering the pathogenesis of these disorders and developing targeted therapies. Continued research in flies holds great promise for advancing our understanding and treatment of rare and ultra-rare diseases.

## Figures and Tables

**Figure 1 insects-15-00870-f001:**
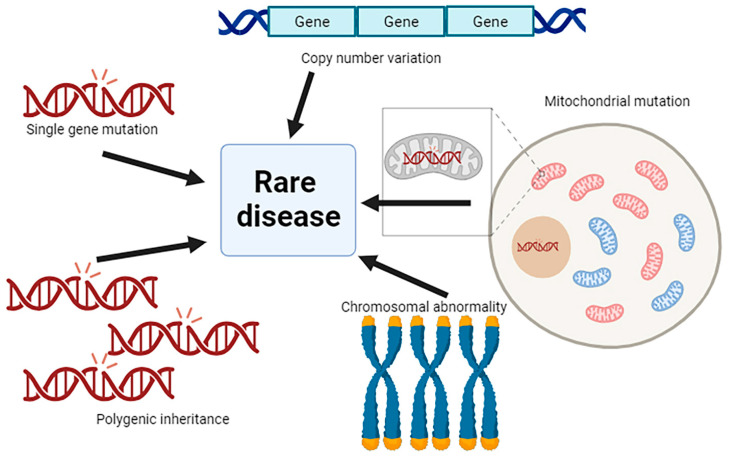
Schematic representation of the genetic alterations that cause rare diseases. Single gene mutations cause genetic aberrations in the DNA sequence of a specific gene. Copy number variation (CNV): the sequences of the genome are repeated. Mitochondrial mutations, the DNA contained in the mitochondria (mtDNA) is mutated. Chromosomal abnormality, the morphology or the number of chromosomes is altered. Polygenic inheritance, more than one gene is mutated. Image generated in BioRender.com (10 October 2024).

**Figure 2 insects-15-00870-f002:**
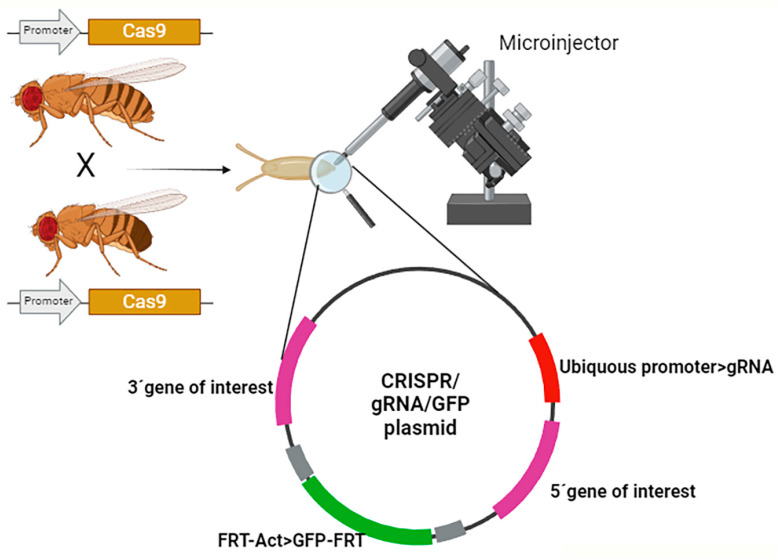
CRISPR/Cas9 system used in *Drosophila* to generate genetic avatars. Representation of CRISPR/Cas9 system to generate mutants in F1 generation; after the cross of parental lines expressing *Cas9* in the germ line, embryos are injected with a plasmid containing the required tools to induce the excision of a region of DNA (exonuclease activity of Cas9), and the re-insertion of the mutated form of the same piece of DNA (endonuclease activity of Cas9). The resulting combination produces the substitution of endogenous exons by mutated exons that reproduce the mutations found in patients. In addition, the plasmid carries a GFP under the control of a constitutive promoter (Actin) to identify the flies that undergo CRIPSR/Cas9 substitution. This GFP cDNA is flanked by two FRT siter to be excised if required in an additional cross with flies that express *flipase*. Image generated in BioRender.com (accessed on 10 October 2024).

## Data Availability

No new data was created in this study.

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
