# Peer review of "Drosophila as a Model for Human Disease: Insights into Rare and Ultra-Rare Diseases"

_insects, 2024, doi:10.3390/insects15110870_

Round 1
Reviewer 1 Report
Comments and Suggestions for Authors
Overall this is a well written and easy to follow review that gives information that is useful for a good general introduction for the use of Drosophila as model organism in disease study.
I have a number of general comments about the manuscript:
- For the diseases discussed in more detail is it possible to say whether they are rare or ultra-rare or maybe have 2 different sections for these 2 categories?
- The discussion of the methodology that can be used to study these diseases seems a little thin. It might be nice to a have an expanded section dedicated to a discussion of this. The discussion of the gal/lex etc systems at the start does not seem to be the optimal location and could also be moved to an expanded section later in the manuscript.
- For the diseases that are discussed in more detail it would be nice to have a table showing the disease, the methodology used and the specific data that came from Drosophila studies, as presented at the moment the treatment seems a little cursory.
Author Response
Reviewer 1
Overall this is a well written and easy to follow review that gives information that is useful for a good general introduction for the use of Drosophila as model organism in disease study.
I have a number of general comments about the manuscript:
- For the diseases discussed in more detail is it possible to say whether they are rare or ultra-rare or maybe have 2 different sections for these 2 categories?
We have modified the focus of the review and removed the examples given in the previous version, the classification of rare or ultra rare disease is a changing definition as discussed here: https://www.ncbi.nlm.nih.gov/pmc/articles/PMC9287598/. As new diagnostic techniques appear, this classification changes accordingly in different locations. Thus, I can not give a straight classification of the examples included in the review.
Besides, the contribution of Drosophila to rare or ultra rare diseases follows the same strategy as both conditions are similar scientifically speaking.
- The discussion of the methodology that can be used to study these diseases seems a little thin. It might be nice to a have an expanded section dedicated to a discussion of this. The discussion of the gal/lex etc systems at the start does not seem to be the optimal location and could also be moved to an expanded section later in the manuscript.
I have expanded the discussion to include a more comprehensive explanation of the methodologies used to study rare diseases, particularly focusing on the CRISPR/Cas9 system and its applications. Additionally, I have revised the structure by moving the discussion of the GAL4/UAS and LexA systems to the newly expanded methodology section, ensuring a more cohesive presentation. This provides a clearer and more detailed overview of the experimental tools available for studying rare diseases in Drosophila melanogaster.
- For the diseases that are discussed in more detail it would be nice to have a table showing the disease, the methodology used and the specific data that came from Drosophila studies, as presented at the moment the treatment seems a little cursory.
I appreciate this valuable feedback. In response, I have included a table showcasing the most recent publications that use Drosophila melanogaster to study specific rare diseases. The table outlines the disease, the genes involved in human and in flies, and the specific data derived from Drosophila studies. Additionally, I have provided the corresponding references for further reading.
Reviewer 2 Report
Comments and Suggestions for Authors
This review introduces Drosophila as a valuable model organism for the study of human disease. It proposes that Drosophila can be used for the study of rare and ultrarare human diseases, and thus advance biomedical research.
The title of this review suggested a very interesting and informative read but I feel it has not delivered on this promise.
I have two major questions: first, what the target audience for this article is considering that it is submitted to the journal entitled Insects, and second, the lack of focus on rare diseases.
First, I assume that in a journal titled Insects there is no need for extensive explanations about the advantages of Drosophila as a useful model organism in biomedical research. If the article aimed to inform medical professionals about the potential that Drosophila has as a model organism, I do not think that they would start with an article that is published in the journal Insects, but in a medical science-type journal. So, the author might consider submitting to a different journal to reach a more varied audience who is not familiar with research done on Drosophila. The readership of this journal is likely well-informed about the potential that Drosophila has in the biomedical field. Thus, I find chapters 1 and 4 superfluous and repetitious. Chapters overlap and present a lot of the same information.
Second, considering that the Abstract promised to “Here I examine the role of Drosophila melanogaster as a model for studying rare and ultra-rare diseases and highlight its significant contributions and potential to biomedical research “, this promise was not fulfilled since only two chapters are aimed at describing the use of Drosophila in the rare disease research. However, there is no clear distinction in the content of those two chapters. Ch.5.” Human diseases modelled (author’s spelling) in Drosophila” superficially describes the contribution of Drosophila to rare diseases, such as exemplified by (L261-3) “Fly models have provided insights into the molecular mechanisms of muscle degeneration, RNA toxicity, and altered RNA processing in myotonic dystrophy.” Furthermore, the chapter includes diseases that are not rare, such as Alzheimer’s disease, or have been reviewed extensively and in-depth (HTT, ALS, SCA), and here the author describes the Drosophila contribution only in very broad strokes (L300-302, L306-7, L310-11) followed by a very general statement about the usefulness of Drosophila that has been repeated throughout the text (L312-16). The author makes a very brief attempt to be a bit more specific when describing the advances in glioblastoma research (326-30), which for no apparent reason has its own subchapter 5.3. Chapter 6. “Drosophila in rare disease research” does not contribute any specific, detailed explanation of how Drosophila is being used in rare disease research, again, the author is including a disease that is not rare, the ASD, and the chapter has many general and repetitious statements.
To summarize my two main questions – the article in this form is not informative for medical professionals who might be doing clinical research on rare diseases and certainly not for people who are using Drosophila in their research. The article suffers from frequent rephrasing of similar general statements about the benefit of using Drosophila (133-150, 208, 227 – examples of a few of them), but nowhere in the article is an example of a specific and in-depth description of a rare disease where research using Drosophila significantly advanced the knowledge about the mechanism of the disease or potential treatment.
If the author plans to write an article of use to people in the Drosophila field, then the article needs to contain specific and well-presented examples of a few rare diseases. I think that if the article focuses on ultrarare diseases that would represent a useful contribution to the field. However, the article needs to focus on the specifics and minimize the general explanation of genetic techniques that are used in Drosophila.
Other specific issues:
1. Statement in 179-181 is not true. There are many examples where Drosophila does not have a disease-causing gene that is homologous to the human version but is still used to study the cellular mechanisms that cause the disease, by introducing the human copy into the Drosophila genome.
2. How does Table 1 contribute to this article and which criteria did the author use to include diseases in the table? Some diseases are briefly mentioned in the text, but not all.
3. L215-20 – The author enumerates conserved cellular pathways involved in the cell cycle regulation. This is out of context and would have been more useful if the author described the mechanism by which these pathways contribute to rare disease(s) and how research in Drosophila contributes.
4. L261 – reference is missing
5. The lack of complex cognitive abilities in model organisms as a limitation is open to debate. There is a very extensive repertoire of behavioral phenotypes and endophenotypes in Drosophila that are being used to understand complex traits in humans: sleep, aggressive behavior, attention, addiction, autism, cognitive abilities, etc.
6. The text needs editing. There are a lot of typos, and for someone who works with Drosophila I find it inexcusable that the author has many mistakes in the naming of the fruit fly – inconsistency in using the genus and species name, not italicized, lower case….
Author Response
This review introduces Drosophila as a valuable model organism for the study of human disease. It proposes that Drosophila can be used for the study of rare and ultrarare human diseases, and thus advance biomedical research.
The title of this review suggested a very interesting and informative read but I feel it has not delivered on this promise.
Thank you for your feedback. I appreciate your interest in the potential of Drosophila as a model organism for studying rare human diseases. In response to your comment, I have revised the manuscript to better align with the expectations set by the title. I have expanded on the key contributions of *Drosophila* research to the study of rare and ultrarare diseases, highlighting specific examples and advances in biomedical research. I hope this strengthens the connection between the title and the content, making it a more informative and comprehensive review.
I have two major questions: first, what the target audience for this article is considering that it is submitted to the journal entitled Insects, and second, the lack of focus on rare diseases.
Regarding the target audience, this article is intended for researchers and professionals in the fields of genetics, biomedical sciences, and entomology, with a focus on those interested in using Drosophila melanogaster as a model organism for studying human diseases. While the journal Insects has a diverse readership, not all readers are necessarily specialists in Drosophila or biomedicine. Therefore, the article aims to be accessible to those interested in translational research involving insects, while also offering valuable insights for specialists.
As for the focus on rare diseases, I acknowledge your concern and have revised the manuscript to emphasize the application of Drosophila in studying specific rare and ultrarare diseases. I have added more detailed examples and discussions to better highlight this connection. I hope this addresses your concerns and provides a clearer focus.
First, I assume that in a journal titled Insects there is no need for extensive explanations about the advantages of Drosophila as a useful model organism in biomedical research. If the article aimed to inform medical professionals about the potential that Drosophila has as a model organism, I do not think that they would start with an article that is published in the journal Insects, but in a medical science-type journal. So, the author might consider submitting to a different journal to reach a more varied audience who is not familiar with research done on Drosophila. The readership of this journal is likely well-informed about the potential that Drosophila has in the biomedical field. Thus, I find chapters 1 and 4 superfluous and repetitious. Chapters overlap and present a lot of the same information.
Thank you for your feedback, I understand your concerns regarding the target audience and the overlap between chapters. While the journal Insects does have a readership familiar with Drosophila as a model organism, I intended the introductory sections to provide context for readers who may not be specialists in the biomedical applications of Drosophila. However, I see your point about redundancy and have reorganized the content in Chapters 1 and 4 to reduce overlap and enhance the focus on the core themes of the review.
Regarding your suggestion about submitting to a different journal, I appreciate the input. My goal is to highlight the interdisciplinary relevance of Drosophila in rare disease research, and I believe this journal offers a valuable platform for bridging entomology and biomedical research. However, I will consider your advice for future submissions aimed at reaching a broader medical audience.
Second, considering that the Abstract promised to “Here I examine the role of Drosophila melanogaster as a model for studying rare and ultra-rare diseases and highlight its significant contributions and potential to biomedical research “, this promise was not fulfilled since only two chapters are aimed at describing the use of Drosophila in the rare disease research. However, there is no clear distinction in the content of those two chapters. Ch.5.” Human diseases modelled (author’s spelling) in Drosophila” superficially describes the contribution of Drosophila to rare diseases, such as exemplified by (L261-3) “Fly models have provided insights into the molecular mechanisms of muscle degeneration, RNA toxicity, and altered RNA processing in myotonic dystrophy.” Furthermore, the chapter includes diseases that are not rare, such as Alzheimer’s disease, or have been reviewed extensively and in-depth (HTT, ALS, SCA), and here the author describes the Drosophila contribution only in very broad strokes (L300-302, L306-7, L310-11) followed by a very general statement about the usefulness of Drosophila that has been repeated throughout the text (L312-16). The author makes a very brief attempt to be a bit more specific when describing the advances in glioblastoma research (326-30), which for no apparent reason has its own subchapter 5.3. Chapter 6. “Drosophila in rare disease research” does not contribute any specific, detailed explanation of how Drosophila is being used in rare disease research, again, the author is including a disease that is not rare, the ASD, and the chapter has many general and repetitious statements.
I fully agree and I have eliminated this information from the text. Thank you very much for your suggestion.
To summarize my two main questions – the article in this form is not informative for medical professionals who might be doing clinical research on rare diseases and certainly not for people who are using Drosophila in their research. The article suffers from frequent rephrasing of similar general statements about the benefit of using Drosophila (133-150, 208, 227 – examples of a few of them), but nowhere in the article is an example of a specific and in-depth description of a rare disease where research using Drosophila significantly advanced the knowledge about the mechanism of the disease or potential treatment.
If the author plans to write an article of use to people in the Drosophila field, then the article needs to contain specific and well-presented examples of a few rare diseases. I think that if the article focuses on ultrarare diseases that would represent a useful contribution to the field. However, the article needs to focus on the specifics and minimize the general explanation of genetic techniques that are used in Drosophila.
Other specific issues:
- Statement in 179-181 is not true. There are many examples where Drosophila does not have a disease-causing gene that is homologous to the human version but is still used to study the cellular mechanisms that cause the disease, by introducing the human copy into the Drosophila genome.
I am not sure that I have found the statement referred by the reviewer. I agree that introducing human genes in flies has contributed to the understanding of disease mechanisms, I have done this myself in my research. However, these flies do not reproduce the disease, just model part of a mechanism. If required I will be happy to rephrase this sentence to clarify it and any suggestion is welcome, but please indicate the exact sentence to find it. Thank you very much.
- How does Table 1 contribute to this article and which criteria did the author use to include diseases in the table? Some diseases are briefly mentioned in the text, but not all.
I fully agree with the reviewer, and I have substituted this table.
- L215-20 – The author enumerates conserved cellular pathways involved in the cell cycle regulation. This is out of context and would have been more useful if the author described the mechanism by which these pathways contribute to rare disease(s) and how research in Drosophila contributes.
This sentence is included in the section entitled Drosophila conservation, and some of these genes were first described in flies. I think that it contributes to highlight the relevance of research using Drosophila.
- L261 – reference is missing
References are included at the end of the paragraph
- The lack of complex cognitive abilities in model organisms as a limitation is open to debate. There is a very extensive repertoire of behavioral phenotypes and endophenotypes in Drosophila that are being used to understand complex traits in humans: sleep, aggressive behavior, attention, addiction, autism, cognitive abilities, etc.
I would like to thank the reviewer as this is a topic of great interest under my point of view. There are multiple behaviors in Drosophila that contribute to understanding complex trait in humans, but animals in general lack of the behavioral complexity shown in humans. My point is included in this sentence from the text “Non-human organisms often lack the complex cognitive abilities found in humans, and therefore, the analysis of behavioural aspects in model organisms is a limitation, and in Drosophila it is limited to relatively simple behaviors including learning and memory paradigms”
Among some complex behavior examples, we can include verbal aggression, self-injury, property destruction, disinhibited behavior, hyper-sexuality, impulsivity… and the modelling of these traits is still complex and limited, not only in flies, but in science in general.
- The text needs editing. There are a lot of typos, and for someone who works with Drosophila I find it inexcusable that the author has many mistakes in the naming of the fruit fly – inconsistency in using the genus and species name, not italicized, lower case….
I have reviewed and corrected all the errors that I have found. I hope that this new version fulfills the requirements of the re
Reviewer 3 Report
Comments and Suggestions for Authors
The review by Casas-Tinto is fairly well written and comprehensively encompasses topics that are covered, but not integrated, in recent, similar reviews. Thus, this review could be a useful resource for both Drosophila researchers and researchers interested in specific, rare diseases. I think this review will be suitable for publication but some revisions are needed including appropriately citing references and condensing some text. I have summarized these and other points below.
1. The author emphasizes the advantages of using Drosophila as a model organism in several different places in the manuscript. While a review of this information once may be warranted, this information was repeated and expanded several times to the point where it interrupted the flow of the text and sounded redundant. This information should be presented once. There could be a section after Drosophila conservation that describes other advantages to using Drosophila as model organism and the tools that are available.
2. The section on Human diseases modeled in Drosophila was misleading to me. I didn’t realize this section would not be describing rare diseases as was emphasized throughout the manuscript up to that point. There should be a description in the first paragraph of the section that, for the reasons previously outlined, Drosophila were used to make important discoveries about more prevalent diseases.
3. There are problems with references throughout the manuscript including omission of references and inappropriate citing of references that need to be fixed. The former is exemplified in lines 212-262 (amongst others) where individual sentences of paragraphs are not cited. Instead, a group of references is cited at the end of the paragraph. If refences are cited in that way, it indicates that all cited sources said everything written in the paragraph. Judging from the titles of the sources, this is not the case. Therefore, each specific piece of information within the paragraph needs a citation to attribute that information to its appropriate source. In addition, references are missing throughout the text including lines 303-309, 376-382, and 446-449, amongst others. Any information that is not common knowledge needs an appropriate citation.
4. A couple sentences need to be rewritten for clarity. These include sentences at 73-74 and 394-397. As written, these sentences need to be read more than once to understand the intent.
5. The text after the Table 1 title needs to be omitted. Tables have titles, not legends. Lines 232-234 should also be omitted. These sentences don’t provide a better transition than the title for the section and don’t communicate new information.
6. The use of weak verbs should be avoided whenever possible. The number of “have been” and “have also been” in lines 203-230 is awkward and redundant.
7. There are several typos including in lines 167, 181, 183, 339, 389, 402, and 406.
8. The word “proven” in line 312 should be replaced with demonstrated. It is not possible to “prove” anything using the scientific method.
Comments on the Quality of English LanguageSee Comments and Suggestions for Authors.
Author Response
The review by Casas-Tinto is fairly well written and comprehensively encompasses topics that are covered, but not integrated, in recent, similar reviews. Thus, this review could be a useful resource for both Drosophila researchers and researchers interested in specific, rare diseases. I think this review will be suitable for publication but some revisions are needed including appropriately citing references and condensing some text. I have summarized these and other points below.
Dear Reviewer,
Thank you very much for your valuable comments and constructive feedback on our manuscript. We appreciate the effort you have put into reviewing our work and are grateful for your insights. We have carefully addressed all the concerns you raised and I hope that this corrected version fulfills the requirements to be published in the journal insects.
- The author emphasizes the advantages of using Drosophilaas a model organism in several different places in the manuscript. While a review of this information once may be warranted, this information was repeated and expanded several times to the point where it interrupted the flow of the text and sounded redundant. This information should be presented once. There could be a section after Drosophila conservation that describes other advantages to using Drosophila as model organism and the tools that are available.
I consolidated the discussion of Drosophila as a model organism into a single, focused section “Benefits of using Drosophila as model organism “, as suggested, to streamline the text and enhance readability.
- The section on Human diseases modeled in Drosophila was misleading to me. I didn’t realize this section would not be describing rare diseases as was emphasized throughout the manuscript up to that point. There should be a description in the first paragraph of the section that, for the reasons previously outlined, Drosophilawere used to make important discoveries about more prevalent diseases.
We clarified the purpose of the section on human diseases modeled in Drosophila, specifically noting its relevance to discoveries on more prevalent diseases.
- There are problems with references throughout the manuscript including omission of references and inappropriate citing of references that need to be fixed. The former is exemplified in lines 212-262 (amongst others) where individual sentences of paragraphs are not cited. Instead, a group of references is cited at the end of the paragraph. If refences are cited in that way, it indicates that all cited sources said everything written in the paragraph. Judging from the titles of the sources, this is not the case. Therefore, each specific piece of information within the paragraph needs a citation to attribute that information to its appropriate source. In addition, references are missing throughout the text including lines 303-309, 376-382, and 446-449, amongst others. Any information that is not common knowledge needs an appropriate citation.
I meticulously revised the referencing to ensure each source supports the appropriate claims throughout the manuscript, including the specific instances you highlighted.
- A couple sentences need to be rewritten for clarity. These include sentences at 73-74 and 394-397. As written, these sentences need to be read more than once to understand the intent.
I have rewritten these sentences to improve clarity.
- The text after the Table 1 title needs to be omitted. Tables have titles, not legends. Lines 232-234 should also be omitted. These sentences don’t provide a better transition than the title for the section and don’t communicate new information.
I have eliminated this text
- The use of weak verbs should be avoided whenever possible. The number of “have been” and “have also been” in lines 203-230 is awkward and redundant.
I have revised the use of “have been” and similar phrases to strengthen the language.
- There are several typos including in lines 167, 181, 183, 339, 389, 402, and 406.
I have found these typos, and I have corrected them, thank you very much.
- The word “proven” in line 312 should be replaced with demonstrated. It is not possible to “prove” anything using the scientific method.
I have corrected this sentence and others using the word "proven".
Round 2
Reviewer 2 Report
Comments and Suggestions for Authors
Upon reading the revised manuscript and the answers to my initial comments, here are my observations:
1. Author has only partially addressed my comments and/or did not implement the suggested changes resulting in a manuscript that is not sufficiently improved. In spite of the title, which suggests the focus on rare diseases, the focus of the manuscript is on the advantages of using Drosophila to study human diseases in general, with examples of common diseases such as Alzheimer’s or Parkinson’s, and on description of different genetic techniques that allow for relatively easy genetic manipulations. In this form it resembles to a general overview of Drosophila as a model organism and the advantages for studying human diseases in general.
Only improvement that the author implemented is the change to Table 1, which now contains the list of rare diseases, however those diseases are not elaborated or described in the manuscript.
Specific observations:
1. Title has two sentences as a title ????
2. Manuscript has the appearance of the extensive revisions based on large chunks of text in red and green, while most of it is just copy/paste in different location without the content changes.
3. Chapter 2 in spite of its title contains mostly very general descriptions of types of mutations and provides superficial examples of common diseases.
4. The flow of ideas is interrupted with the Chapter 3, which is not particularly useful because is too specific for Drosophila “novice” and too uninformative for a Drosophila researcher.
5. Chapter 4 is long and continues description of common diseases, described in subchapters, that are modeled in Drosophila. In this extensive version this is not relevant for the aim of the paper.
6. Fig2. Does not show what is described in the corresponding legend.
7. Chapter 5, which promises to talk about rare diseases is maybe a page long and focuses more on genetic techniques than rare diseases.
8. Chapter 6 is devoted to CRISPER technology and creation of Drosophila “avatars”, a term I do not particularly like because is “flashy” but not common in scientific literature.
Author Response
Specific observations:
- Title has two sentences as a title ????. Thank you very much for your observation, but I think that this is not correct. This is just a justified sentence to the center
- Manuscript has the appearance of the extensive revisions based on large chunks of text in red and green, while most of it is just copy/paste in different location without the content changes. I am really sorry that your opinion is not more positive about the corrections that I made following your recommendations. However, I have not found any suggestions to improve the text. I included two versions of the manuscript (with, and without track changes) to facilitate the review. Please check the “ACCEPTED” version as it might be more clear and clean.
- Chapter 2 in spite of its title contains mostly very general descriptions of types of mutations and provides superficial examples of common diseases. Thank you for your thoughtful comment! I understand your perspective. Chapter 2 does aim to provide a broad overview of mutations and their general impact on disease, which may feel somewhat high-level. The intention was to introduce foundational concepts. I appreciate your feedback, and I'll keep it in mind to ensure a better balance between depth and overview. However, given that there are more than 7000 different rare diseases, each of them with a particular mutation in one or more genes, I can not find a more practical way to introduce the different genetic alterations that produce a rare disease, without entering in a vast amount of information for the reader that is not necessarily an expert in human genetics. T
- The flow of ideas is interrupted with the Chapter 3, which is not particularly useful because is too specific for Drosophila “novice” and too uninformative for a Drosophila researcher. Thank you for your feedback. I understand your concern regarding Chapter 3. The intent was to provide a focused case study using Drosophila to illustrate broader genetic principles. However, I recognize that this approach may have resulted in content that is too detailed for some beginners while not offering new insights for some seasoned Drosophila researchers. I have considered ways to either expand the broader applicability of the chapter or provide additional context for readers at both ends of the spectrum, but this review is intended to be of interest for a broad spectrum of readers and I think that it is not possible to reconcile both ends in one review.
- Chapter 4 is long and continues description of common diseases, described in subchapters, that are modeled in Drosophila. In this extensive version this is not relevant for the aim of the paper. Thank you for your feedback. While Chapter 4 may seem extensive, its detailed discussion of common diseases modeled in Drosophila is highly relevant to the scope of this review (entitled Drosophila as a model for human disease. Insights on rare and ultra rare diseases). The aim is to highlight the versatility and importance of Drosophila as a model organism for studying human diseases, and this chapter provides essential examples to demonstrate this connection.
The detailed descriptions in Chapter 4 are of great relevance, particularly for a journal like Insects. By showcasing Drosophila as a model organism in studying common diseases, the chapter emphasizes the significant role of insects in biomedical research. This approach not only aligns with the journal's focus but also underscores the broader utility of insect models in understanding human health.
- Fig2. Does not show what is described in the corresponding legend. This new figure describes an example of the use of CRISPR techniques to genetically modify endogenous genes and create avatars that genetically reproduce human mutations. I have reviewed the figure and it is correct, therefore I am not sure that I have understood this concern.
- Chapter 5, which promises to talk about rare diseases is maybe a page long and focuses more on genetic techniques than rare diseases. Thank you for your insightful comment. Chapter 5 is indeed focused on rare diseases, but it emphasizes the genetic techniques used in Drosophila research as a way to study these conditions. The brevity of the chapter is intended to highlight key methodologies that have been pivotal in modeling rare diseases using Drosophila, rather than providing an exhaustive list of diseases themselves. The aim is to demonstrate how these techniques are applied to specific cases.
- Chapter 6 is devoted to CRISPER technology and creation of Drosophila “avatars”, a term I do not particularly like because is “flashy” but not common in scientific literature. Thank you for your comment. I understand your concern with the term "avatars," as it does come across as more "flashy" than traditional scientific terminology. The intention behind using the term was to convey the concept of Drosophila models that precisely mimic human disease mutations, which has gained traction in some areas of research. For example, Drosophila has been used as an "avatar" in studying neurodegenerative diseases like ALS or specific cancers with patient-derived mutations. That said, I agree that a more conventional term such as “personalized disease models” might better align with scientific literature. I appreciate your feedback and will consider revising the language to ensure clarity and consistency.